# Quantum dynamics in 1D lattice models with synthetic horizons

Corentin Morice[1], Dmitry Chernyavsky[2], Jasper van Wezel[1],
Jeroen van den Brink[2,3] and Ali G. Moghaddam[2,4]

**1** Institute for Theoretical Physics and Delta Institute for Theoretical Physics,
University of Amsterdam, 1090 GL Amsterdam, The Netherlands
**2** Institute for Theoretical Solid State Physics, IFW Dresden,
Helmholtzstr. 20, 01069 Dresden, Germany
**3** Institute for Theoretical Physics and Würzburg-Dresden Cluster of Excellence ct.qmat,
Technische Universität Dresden, 01069 Dresden, Germany
**4** Department of Physics, Institute for Advanced Studies in Basic Sciences (IASBS),
Zanjan 45137-66731, Iran

## Abstract

We investigate the wave packet dynamics and eigenstate localization in recently proposed generalized lattice models whose low-energy dynamics mimics a quantum field theory in (1+1)D curved spacetime with the aim of creating systems analogous to black holes. We identify a critical slowdown of zero-energy wave packets in a family of 1D tight-binding models with power-law variation of the hopping parameter, indicating the presence of a horizon. Remarkably, wave packets with non-zero energies bounce back and reverse direction before reaching the horizon. We additionally observe a power-law localization of all eigenstates, each bordering a region of exponential suppression. These forbidden regions dictate the closest possible approach to the horizon of states with any given energy. These numerical findings are supported by a semiclassical description of the wave packet trajectories, which are shown to coincide with the geodesics expected for the effective metric emerging from the considered lattice models in the continuum limit.


## Contents



# 1 Introduction

Connections between different fields of physics have proven fruitful by opening entirely new research avenues in recent years. Dualities between gravitational and many-body theories have become important tools in the study of quantum critical systems [1], while the search for electronic states following the Dirac equation, prevalent in high-energy physics, led to the discovery of topological insulators, one of the most major fields in condensed matter physics in decades. In this context, replicating some of the physics of curved spacetime in condensed matter settings has been proposed as a promising way of on the one hand understanding gravitational problems in a simplified setting, and on the other of searching for novel, gravity-inspired, physical effects within condensed matter theory [2]. In particular, in a seminal work Unruh proposed the construction of an analog black hole horizon and its radiation using a fluid flowing with a spatially varying speed profile that is partly subsonic and partly supersonic [3,4]. Similarly, many proposals for analog gravity setups emulating a broad range of emergent curved spacetimes have been put forward in a variety of electronic, acoustic, optical and even magnetic and superconducting settings [2, 5–26]. Some of these proposals have already been implemented in experiments, mainly using Bose-Einstein condensates [27–29]. In all of these proposed and realised black hole analogues however, the role played by the atomic lattice (periodic or otherwise) remains largely unexplored, even though it is an essential component of any condensed matter system.

Recently, it was shown that a black hole analogue may be realized in Weyl semimetals (WSMs) by tilting the Weyl cone as a function of real space, transitioning from a type-I to a type-II WSMs [30–37]. The tilt causes part of the band structure close to the Weyl node to become progressively flatter as the type I-type II transition point is approached. This is a direct analogy for the tilting of a light cone close to a black hole, in which case the surface on which the light cone tips across the time axis defines an event horizon, beyond which all light is trapped [30]. It has been shown that $Zn_2In_2S_5$ sits precisely at the transition, and was coined to be a 'type-III' Weyl semimetal [34]. Tuning the tilt of the cone across real space could be achieved using structural distortions, spin textures, or external position-dependent driving [32, 38–42].

To circumvent the difficulty of defining spatial variations in the tilt of Weyl cones, which themselves require reciprocal space and translational symmetry to be defined, previous studies

generally assume that the control parameter responsible for tilting changes smoothly, and that a band structure varying as a function of real space can be defined, despite the absence of translation symmetry. Here, we take a more rigorous approach, and define a Hamiltonian in real space with a hopping that varies as a function of position. This has the benefit of allowing us to explicitly study new effects that arise from the presence of the lattice and the potential limitations it imposes on the dynamics in the emergent analogue gravitational system.

In a recent paper, we considered generalized nearest-neighbor tight-binding (TB) models with position-dependent hopping and showed that their low-energy dynamics is similar to that of a Dirac field with a position-dependent velocity, which mimics the presence of a background curved spacetime [43]. Rather than the Dirac equation and Weyl nodes in 3D we considered single band systems in 1D with progressive band-flattening in real space which are fully tractable numerically, allowing direct comparison with analytical semi-classical treatments of the problem.

Here, we investigate how and when the power-law position-dependent TB models introduced in [43] yield analogues for horizon physics, as witnessed by the critical slowing down of wave packet dynamics. We derive rigorously the generalized version of the semiclassical equations presented in [43], and find a formal solution for semiclassical trajectories in the most general case, accompanied by explicit solutions for the power-law dependencies to compare with the numerical results. Furthermore, we develop an analytical approach to solving these models and show that all the eigenstates in the models with power-law variation of the hopping are also localized in a power-law manner. Consistent with the observed wave packet dynamics, the low-lying states localize on the horizon, whereas high-energy states are exponentially suppressed in a region near the horizon.

In the following, we first introduce the general model and its low-energy sector with gravitational analogies (Sec. 2). Then, we calculate numerically the time evolution of wave packets for power-law hopping models in Sec. 3. We derive semiclassical equations of motion along with their solutions and compare them to numerical results in Sec. 4. This is followed by a discussion of the eigenstates in Sec. 5, before we conclude in Sec. 6.

## 2 TB Models and their gravitational analogy

Consider electrons on a one-dimensional lattice of $N$ sites, with nearest-neighbour hopping only:

$$\hat{\mathcal{H}} = -\sum_{n=1}^{N-1} t_n \left( \hat{a}_n^\dagger \hat{a}_{n+1} + \hat{a}_{n+1}^\dagger \hat{a}_n \right). \tag{1}$$

Here, $t_n$ is a position-dependent hopping parameter whose amplitude increases with $n$. This Hamiltonian has a particle-hole symmetry (PHS) represented by the transformation [44,45]

$$\begin{aligned}
\hat{a}_n &\to \hat{\mathcal{C}} \hat{a}_n \hat{\mathcal{C}}^{-1} = (-1)^n \hat{a}_n^\dagger, \\
\hat{a}_n^\dagger &\to \hat{\mathcal{C}} \hat{a}_n^\dagger \hat{\mathcal{C}}^{-1} = (-1)^n \hat{a}_n,
\end{aligned} \tag{2}$$

under which the second-quantized Hamiltonian remains invariant: $\hat{\mathcal{C}} \hat{\mathcal{H}} \hat{\mathcal{C}}^{-1} = \hat{\mathcal{H}}$. The PHS can also be seen explicitly for the corresponding first-quantized Hamiltonian $\hat{H} = -\sum_{n=1}^{N-1} t_n |n\rangle\langle n+1| + h.c.$ which anticommutes with the PHS operator $\hat{P} = \hat{U}\hat{K}$ consisting of complex conjugation $\hat{K}$ and the unitary operator $\hat{U} = \sum_{n=1}^N (-1)^n |n\rangle\langle n|$

$$\hat{U} = \sum_{n=1}^N (-1)^n |n\rangle\langle n|. \tag{3}$$

We see that $\{\hat{U}, \hat{H}\} = \{\hat{P}, \hat{H}\} = 0$ since the Hamiltonian contains real elements only. We note that the presence of $\hat{K}$ in the definition of any PHS operator is required to ensure its antiunitarity, a property which manifests more clearly by acting with this operator on the wave functions. In particular, considering a uniform hopping where the plane waves $\phi_k(n) = e^{ikn}$ are the eigenstates with energies $\varepsilon_k = -2t \cos k$, we find $\hat{P}\phi_k(n) = e^{i(\pi-k)n} \equiv \phi_{\pi-k}(n)$, with energy opposite to the original state.

The particle-hole symmetry of this model is analogous to charge-conjugation symmetry in quantum electrodynamics. Therefore, the presence of this symmetry already suggests the possibility of emerging relativistic aspects, and in particular of a Dirac picture for its low-energy excitations. On the other hand, the PHS can be broken by adding a potential energy, thus we concentrate on models with PHS throughout this work.

## 2.1 Low-energy limit

The low-energy properties of these lattice models mimic those of a Dirac field in curved space-time. This is made plausible by the observation that approximating the full lattice by disconnected (periodic) sections, results in local band structures $\varepsilon(n,k) \sim -2t_n \cos k$[1]. Each of these has two Fermi points $k_F = \pm\frac{\pi}{2}$ at half-filling. Accordingly, we can define the local Fermi velocity $v_F(n) = \partial_k \varepsilon(n, k = \pm k_F) \sim \pm 2t_n$. Motivated by this observation, we construct a precise correspondence of the low-energy properties of the lattice model to a Dirac field with position-dependent velocity by introducing the transformation

$$a_n = \sum_{\nu=\pm} \hat{\psi}_\nu(x_n) e^{i\nu k_F n}. \tag{4}$$

We additionally take the continuum limit, in which $x_n \equiv x$ becomes a continuous variable and $\hat{\psi}_\nu(x_{n+1}) \approx \hat{\psi}_\nu(x_n) + \partial_x \hat{\psi}_\nu(x_n)$. The resulting Hamiltonian is

$$\hat{\mathcal{H}} \approx -\int dx \, t(x) \sum_{\nu=\pm} \left[ \hat{\psi}_\nu^\dagger (-i\nu\partial_x) \hat{\psi}_\nu - e^{2i\nu k_F x} \hat{\psi}_\nu^\dagger (-i\nu\partial_x) \hat{\psi}_{-\nu} + h.c. \right]. \tag{5}$$

The second term includes fast oscillations $e^{2i\nu k_F x}$, and can be neglected in the limit of slowly-varying fields $\hat{\psi}_\nu(x)$. Therefore, the lattice model is equivalent in the continuum limit to a model for the 1D massless Dirac field $\hat{\Psi} = (\hat{\psi}_+, \hat{\psi}_-)^T$ governed by

$$\hat{\mathcal{H}}_D = \frac{1}{2} \int dx \left\{ \hat{\Psi}^\dagger \left[ i\sigma_z v(x)\partial_x \right] \hat{\Psi} + h.c. \right\}, \tag{6}$$

with a space-dependent velocity such that $v(x) = 2t(x)$, and Pauli matrix $\sigma_z$ acting on the spinor[2]. By variation we obtain the equation of motion

$$i\partial_\tau \hat{\Psi} = i\sigma_z \left[ v(x)\partial_x + \frac{1}{2}\frac{dv}{dx} \right] \hat{\Psi}, \tag{7}$$

---

[1]Throughout the paper, we consider the lattice constant $a_0 = 1$ and also work in the natural units with $\hbar = 1$.

[2]This decoupling is different from what was done in the literature, e.g. in Ref. [46]. There, in section 3.2, a system with position-dependent Fermi velocity was applied on a spinor using $\sigma_x$, as made clear in Eqs. (11) and (29). In our case, we use $\sigma_z$ and have left movers and right movers that are decoupled —this is the correct physical basis if one wants to consider the reflection/transmission of such movers at the horizon. This difference, at first sight, leads to opposite results: while we will show in the following that there is no transmission through the horizon in our model, Ref. [46] obtains full transmission.

This can be explained by the fact that our situation is a very peculiar case in the formalism of Ref. [46]. The velocity profile in that paper consists of three regions: two with constant velocities ($v_+$ and $v_-$) and a third region in between where the velocity goes linearly from $v_-$ to $v_+$. In addition, it is clear from Eq. (42) that $v_-$ and $v_+$ have to be of the same sign, so one of them has to be set to zero to have a horizon. When either $v_+$ or $v_-$ becomes zero, one cannot use Peres' scattering treatment which is also prohibited by Eq. (42) in the paper, because there will be no state available on that side. Therefore when we have a horizon, his analysis and the result of full transmission is no longer valid.

which is identical to that describing the dynamics of a 1D massless Dirac field in the presence of the background $(1 + 1)$D metric (see Appendix A)

$$ds^2 = -v^2(x)d\tau^2 + dx^2. \tag{8}$$

This metric can possess a horizon at positions where $v(x) = 0$. Based on this property, we mainly consider power-law variations of the hopping integrals, which yields effective local velocities of the form $v(x) = v_0 x^\gamma$. In the following section we will study the wave packet dynamics on lattices with power-law hopping variation and discuss the results in light of their gravitational resemblances.

It is worth noting that by applying certain coordinate transformations, the general metric (8) can be put in other forms which are more familiar in the context of general relativity. For instance, it becomes a generalized Schwarzschild metric

$$ds^2 = -\tilde{v}^2(\xi)\,d\tau^2 + \frac{d\xi^2}{\tilde{v}^2(\xi)}, \tag{9}$$

by choosing a new spatial coordinate $\xi = u(x)$ such that $d\xi/dx = v(x)$ and equating $v(x) = v[u^{-1}(\xi)] \equiv \tilde{v}(\xi)$. Also by apply another temporal transformation

$$d\tau \to d\tau - \frac{\sqrt{1 + \tilde{v}^2(\xi)}}{\tilde{v}^2(\xi)}d\xi, \tag{10}$$

we obtain

$$ds^2 = -\tilde{v}^2(\xi)\,d\tau^2 + 2\sqrt{1 + \tilde{v}^2(\xi)}\,d\xi d\tau - d\xi^2, \tag{11}$$

which, considering the particular case of $\tilde{v}^2(\xi) = \alpha^2\xi^2 - 1$, reduces to the metric studied earlier corresponding to a (1+1)D anti-de Sitter spacetime [43].

## 3 Wave packet dynamics

Motivated by the equivalence between the low-energy dynamics of the position-dependent lattice model and Dirac particles in a curved space, in this part we explore numerically the wave packet dynamics on these lattices. We focus on the cases where the hopping grows as a power-law with position:

$$t_n = \left(\frac{n}{N-1}\right)^\gamma, \qquad n = 1, \cdots, N-1, \tag{12}$$

such that the maximum hopping in the system is equal to one. A key aspect of black hole physics is that an observer at infinity will see a wave packet falling towards a black hole become sharper and slower as it approaches the horizon, before asymptotically reaching the horizon shaped as a Dirac distribution. As we expect the velocity of a wave packet in the lattice model to be associated with the local strength of the hopping, the equivalent of a horizon in the lattice model may occur where the Fermi velocity approaches zero, i.e. in the vicinity of the bond of the lattice between site $n = 1$ and a 'virtual' site $n = 0$.

To compute the time evolution of an initial wave packet in the lattice model we work in the basis of the diagonalized Hamiltonian and use the expression

$$|\psi(\tau)\rangle = \sum_\ell e^{-iE_\ell \tau} |\ell\rangle \langle \ell|\psi(0)\rangle, \tag{13}$$

where $|\ell\rangle$ signifies the $\ell$th eigenvector of $H$, and $\tau$ denotes time. We define a Gaussian wave packet by:

$$\psi_n(\tau = 0) = \frac{1}{\sqrt[4]{\pi w^2}} e^{-\frac{1}{2}\left(\frac{n-n_0}{w}\right)^2} e^{ip_0 \frac{n}{N-1}}, \tag{14}$$
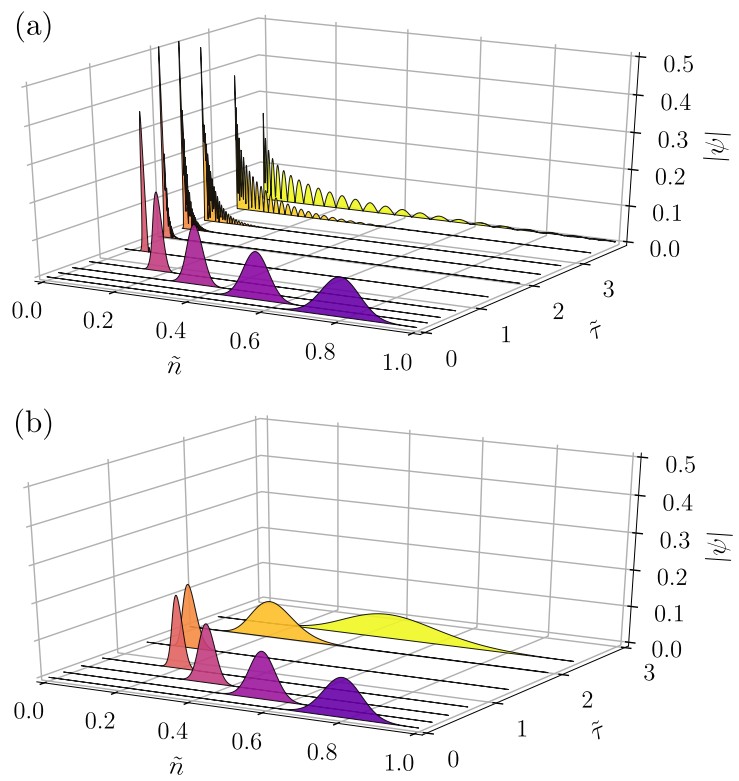

Figure 1: Time evolution of a Gaussian wave packet in the lattice model, with $\gamma = 1$, $\tilde{n}_0 = 0.8$, and $\tilde{w} = 0.05$ for a lattice of size $N = 1001$. The initial momenta are $p_0 = -\pi/2$ (top) and $p_0 = -0.9 \times \pi/2$ (bottom). In the first case, the wave packet slows down and localizes at the origin of the lattice, where it disintegrates. In the second case, it is reflected before reaching the origin.

where $w$ is the width, $n_0$ the initial position and $p_0$ the initial velocity of the wave packet. Note that, for the sake of simplicity, we introduce the rescaled parameters $\tilde{n} = n/(N-1)$, $\tilde{\tau} = \tau/(N-1)$, and $\tilde{w} = w/(N-1)$.

Let us first consider a linearly increasing hopping parameter ($\gamma = 1$). In that case, we find two different possible types of behavior for the wave packet, depending on whether $p_0$ is equal to or different from $-\pi/2$. Example time evolutions of both cases are presented in Fig. 1. In each case, the wave packet starts by sharpening and slowing down as it moves towards $n = 0$. Wave packets with $p_0 \neq -\pi/2$ never reach the origin of the lattice, and instead come to a standstill at non-zero $n$ before moving away from the origin and broadening again. In contrast, the peak position of wave packets with $p_0 = -\pi/2$ continues to approach the origin of the lattice indefinitely. As their peak comes close to $n = 0$, these wave packets start to form ripples in their tails, which move away from the origin. Eventually, the wave packet consists almost entirely of these ripples, but conserves a maximum amplitude at the origin of the lattice.

The observed asymptotic localization of wave packets at the origin coincides with the key feature expected for wave packet dynamics in the presence of a horizon. One key difference with what is expected close to a black hole horizon, however, is the formation of ripples. This feature of the model can be understood as a consequence of the discreteness of the lattice and the unitarity of time evolution. Indeed, consider two different wave packets both with $p_0 = -\pi/2$. If they could asymptotically localise at the origin of the lattice, they would become indistinguishable from each other, and it would then be impossible to propagate them back to their distinct original configurations by reversing time. Since this cannot be the case in our system, which has unitary time evolution, the two wave packets have to develop specific

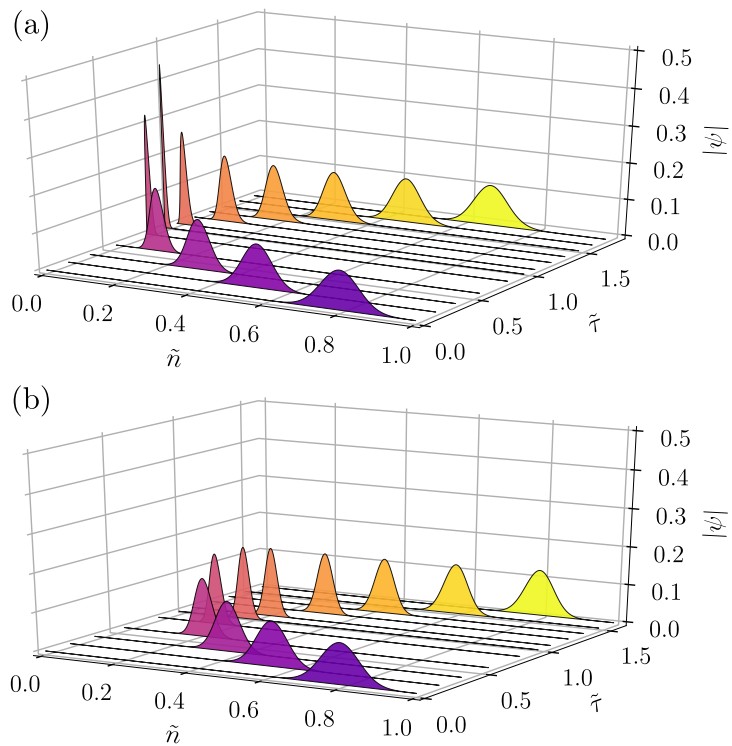

Figure 2: Time evolution of a Gaussian wave packet in the lattice model, with $\gamma = 1/2$, $\tilde{n}_0 = 0.8$, and $\tilde{w} = 0.05$ for a lattice of size $N = 1001$. The initial momenta are $p_0 = -\pi/2$ (top) and $p_0 = -0.7 \times \pi/2$ (bottom). In both cases, the wave packet is reflected, but it only reaches the origin of the lattice in the first case.

features close to the origin of the lattice, which act as signatures of unitary time evolution. These features here come in the shape of ripples, propagating away from the origin of the lattice. Wave packets with $p_0 \neq -\pi/2$ do not exhibit eternal slowdown, pointing to the fact that the horizon physics can only be probed at a critical initial momentum.

We now turn to the case of $\gamma = 1/2$, describing a square-root position-dependence of the hopping. The time evolution of the wave packet amplitude for two values of $p_0$ is displayed in Fig. 2. In this case neither of the wave packets localizes at the zero-velocity point, and both turn around and move out to infinity at late times. Additionally, we don't observe any formation of ripples, unlike in the case $\gamma = 1$. One similarity with this case, however, is that wave packets with $p_0 = -\pi/2$ reach the origin of the lattice, while wave packets with $p_0 \neq -\pi/2$ reverse direction at a nonzero distance from the origin. All these features suggest that while the model with $\gamma = 1$ gives rise to a horizon, the model with $\gamma = 1/2$ does not. In the next section we will combine numerical results with a semiclassical analysis and show that indeed $\gamma = 1$ is a critical value below which the model does not contain any horizon physics. In contrast, for $\gamma \geq 1$, the point $n = 0$ resembles the horizon of a black hole with low-energy particles eternally slowing down upon approaching it.

## 4 Semiclassical dynamics

To further analyze the time evolution of wave packets, it is constructive to compare the numerical results to a semiclassical description for the trajectories of the wave packet center of mass. We introduce a continuous function $\tilde{\psi}(x)$ which coincides with the wave function of

the lattice model at the discrete lattice points, so that $\tilde{\psi}(x_n) = \psi_n$ with $x_n = n/(N-1)$.

For a general position-dependent hopping model, the equation for the energy eigenvalue can be written as the recursive relation

$$\varepsilon\psi_n = -t_{n-1}\psi_{n-1} - t_n\psi_{n+1}. \tag{15}$$

Assuming that we can expand $\tilde{\psi}(x)$ in a Taylor series, we can relate $\psi_{n\pm1}$ to $\tilde{\psi}(x_n)$ exactly, by

$$\psi_{n\pm1} = \tilde{\psi}(x_n \pm \delta x) = \sum_{m=0}^{\infty} \frac{(\pm\delta x)^m}{m!} \frac{d^m\psi(x_n)}{dx^m}$$

$$= \sum_{m=0}^{\infty} \frac{(\pm i\,\delta x)^m}{m!}\, \hat{p}^m\, \tilde{\psi}|_{x_n} = e^{\pm i\delta x\hat{p}}\, \tilde{\psi}(x_n), \tag{16}$$

where $\delta x = 1/(N-1)$, and we replaced derivatives using $\hat{p}^m = (-id/dx)^m$. We also summed over odd and even powers separately, which can be done formally by expressing them sum in terms of sines and cosines of the momentum operator. The result is nothing but the well-known expression of the translation operator $\mathcal{T}_{\Delta x} = e^{-i\Delta x\hat{p}}$. Now assuming $t_n \equiv t(x_n)$ in Eq. (15) with a well-behaved function $t(x)$, the eigenvalue equation can be re-written as

$$i\partial_\tau \tilde{\psi}(x,\tau) = -\left[t(\hat{x} - \delta x)\,e^{-i\delta x\hat{p}} + t(\hat{x})\,e^{i\delta x\hat{p}}\right]\tilde{\psi}(x,\tau), \tag{17}$$

where $\tilde{\psi}(x,\tau) = \tilde{\psi}(x)e^{-i\varepsilon\tau}$. The right-hand side in Eq. (17) can be interpreted as the continuum Hamiltonian

$$\tilde{H} = -e^{-i\delta x\hat{p}}\,t(\hat{x}) - t(\hat{x})\,e^{i\delta x\hat{p}}, \tag{18}$$

where we used the fact that $[t(\hat{x})e^{i\delta x\hat{p}}]^\dagger = e^{-i\delta x\hat{p}}\,t(\hat{x}) = t(\hat{x}-\delta x)e^{-i\delta x\hat{p}}$ to write the Hamiltonian in manifestly Hermitian form. The corresponding Heisenberg equations of motion (EOM) for the momentum and position operators read

$$i\frac{d\hat{x}}{d\tau} = [\hat{x}, \tilde{H}] = -\delta x\left[e^{-i\delta x\hat{p}}\,t(\hat{x}) - t(\hat{x})\,e^{i\delta x\hat{p}}\right], \tag{19}$$

$$i\frac{d\hat{p}}{d\tau} = [\hat{p}, \tilde{H}] = i\left[e^{-i\delta x\hat{p}}\,t'(\hat{x}) + t'(\hat{x})\,e^{i\delta x\hat{p}}\right], \tag{20}$$

with $t'(x) = dt/dx$. Now, neglecting the commutation relations between $\hat{x}$ and $\hat{p}$, we obtain semiclassical EOM for the expectation values $x$ and $p$:

$$\frac{dx}{d\tau} \approx 2t(x)\sin p, \tag{21}$$

$$\frac{dp}{d\tau} \approx 2t'(x)\cos p. \tag{22}$$

Here, we rescaled time and momentum as $\tau \to \tau/\delta x$ and $p \to p/\delta x$. Differentiating Eq. (21) and replacing the derivative of $p$ by the right-hand side of Eq. (22) yields

$$\frac{d^2x}{d\tau^2} \approx 2\frac{d}{dx}t^2(x), \tag{23}$$

which is a straightforward second order differential equation for the dynamics of the position.

## 4.1 General solutions for the trajectories

In this part, we present the formal solution to the semiclassical Eqs. (21) and (22) for general form of the hopping. We first note that defining an auxiliary function $\mathcal{F}(x)$ such that $dx/d\tau = \mathcal{F}[x(\tau)]$, we have

$$\frac{d^2x}{d\tau^2} = \mathcal{F}'[x(\tau)]\frac{dx}{d\tau} = \frac{1}{2}\frac{d}{dx}\mathcal{F}^2\Big|_{x=x(\tau)}. \tag{24}$$

This allows us to write Eq. (23) in the simplified form

$$\frac{d}{dx}\Big[\mathcal{F}^2 - 4t^2(x)\Big] = 0. \tag{25}$$

Therefore, $\mathcal{F}^2 - \big[2t(x)\big]^2$ is just a constant $A$ and replacing $\mathcal{F}$ with its original definition, we end up with the first order differential equation

$$\frac{dx}{d\tau} = \pm\sqrt{\big[2t(x)\big]^2 + A}, \tag{26}$$

with a formal solution

$$\tau = \pm\int_{x_0}^{x}\frac{dx}{\sqrt{\big[2t(x)\big]^2 + A}} + B, \tag{27}$$

for the most general case. The integration constant $A$ can be fixed using Eqs. (21) and (26) at $\tau = 0$. This yields $A = -\big[2t(x_0)\cos p_0\big]^2$, with $x_0$ and $p_0$ indicating the position and momentum at $\tau = 0$. This also fixes the signs in Eqs. (26) and (27) to be $-\mathrm{sgn}[t(x)\sin p]$. Notice that at a turning point $p = 0$ when momentum undergoes a sign change, and at points where the sign of the hopping parameter switches, the sign in Eqs. (26) and (27) also changes. At those points, care should be taked to choose the constant $B$ such that the different parts of the solution match.

Combining Eqs. (21) and (22), we obtain the new equation

$$\frac{dp}{dx} = \frac{t'(x)}{t(x)}\cot p, \tag{28}$$

which directly relates the position and the momentum. It has the solution

$$\cos p = \frac{t(x_0)\cos p_0}{t(x)}. \tag{29}$$

This relation shows that $t(x)\cos p$ is a constant of motion and, in fact, we can assign $E_{\mathrm{wp}} = -2t(x)\cos p$ as the conserved average energy of the wave packet in a semiclassical sense. In particular, we see that for initial value $p_0 = \pm\pi/2$, the momentum of the wave packet remains constant throughout the time evolution. Since it also implies $E_{\mathrm{wp}} = 0$ for all times, this can also be thought of as a consequence of energy conservation. Finally, Eq. (29) determines the position of the turning point of the wave packet (when $p = 0$) and in particular the minimum distance from the horizon, as $t(x_{\min}) = t(x_0)\cos p_0$.

## 4.2 Trajectories for power-law hopping

Although Eqs. (27) and (29) give a general solution for the semiclassical equations, the former is just a formal expression in terms of an integral. Here, we therefore focus on the specific case of power-law hopping, defined as $t(x) = x^\gamma$, for which the trajectories read

$$\tau = \pm\frac{x}{\sqrt{A}}\,{}_2F_1\left(\frac{1}{2},\frac{1}{2\gamma},1+\frac{1}{2\gamma},\frac{4x^{2\gamma}}{-A}\right)\Bigg|_{x_0}^{x} + B, \tag{30}$$

using the hypergeometric function $_2F_1(a, b, c; z)$ with three real parameters $a$, $b$, $c$, and the variable $z$. This expression gives a real value only when $-4x^{2\gamma}/A > 1$ or equivalently $x > x_0(\cos p_0)^{1/\gamma}$, in agreement with the turning point being given by $x_{\min} = x_0(\cos p_0)^{1/\gamma}$.

For the special cases of $\gamma = 1$ or $\gamma = 1/2$, corresponding respectively to linear and square-root forms of position-dependence, the solution of Eq. (30) simplifies to

$$\tau = B \pm \begin{cases} \frac{1}{2}\log\left(\frac{x + \sqrt{x^2 - x_0^2 \cos^2 p_0}}{x_0 + x_0 |\sin p_0|}\right) & \gamma = 1, \\ \sqrt{x - x_0 \cos^2 p_0} - \sqrt{x_0} |\sin p_0| & \gamma = \frac{1}{2}. \end{cases}$$

Inverting this result and writing the position $x$ in terms of the time $\tau$, while also choosing values for $B$ by matching different parts of the solution, yields

$$x_{\gamma=1} = \frac{x_0}{2}\left[(1 + \sin p_0)e^{2\tau} + (1 + \sin p_0)e^{-2\tau}\right], \tag{31}$$

$$x_{\gamma=1/2} = x_0 + 2\tau\sqrt{x_0}\sin p_0 + \tau^2, \tag{32}$$

for linear and square-root position-dependence respectively. Substituting these back into Eq. (29), the evolutions of the corresponding momenta are found to be

$$\cos p_{\gamma=1} = \frac{2\cos p_0}{(1 + \sin p_0)e^{2\tau} + (1 - \sin p_0)e^{-2\tau}} \tag{33}$$

$$\cos p_{\gamma=1/2} = \frac{\sqrt{x_0}\cos p_0}{\sqrt{x_0 + 2\tau\sqrt{x_0}\sin p_0 + \tau^2}}. \tag{34}$$

## 4.3 Zero-energy wave packets and equivalence to geodesics

In the limit of $p_0 = -\pi/2$, the constant $A$ vanishes, and the general spatial trajectory of Eq. (26) becomes

$$\frac{dx}{d\tau} = \pm 2t(x) \equiv \pm v(x). \tag{35}$$

Not surprisingly, the semiclassical trajectories in this limit coincide with the lightlike geodesics ($ds^2 = 0$) of the general metric in Eq. (8). In the case of power-law hopping variations, the integral equation (27) simplifies to

$$\tau = \pm \int_{x_0}^{x} \frac{dx}{2x^\gamma} = \begin{cases} \pm\frac{\left(x^{1-\gamma} - x_0^{1-\gamma}\right)}{2(1-\gamma)} & \gamma \neq 1, \\ \pm\frac{1}{2}\log\left(\frac{x}{x_0}\right) & \gamma = 1, \end{cases} \tag{36}$$

which in turn leads to

$$x = \begin{cases} \left|x_0^{1-\gamma} - 2(1-\gamma)\tau\right|^{\frac{1}{1-\gamma}} & \gamma \neq 1, \\ x_0\exp(-2\tau) & \gamma = 1. \end{cases} \tag{37}$$

Notice that these expressions agree with Eqs. (31) and (34) after substituting $p_0 = -\pi/2$.

## 4.4 Comparing semiclassical and numerical results

Figure 3 shows a comparison between the semiclassical trajectories given by Eq. (37) and numerical calculations of the exact time evolution of the wave packet peak position on the lattice. Both the numerical and semiclassical results show that for $p_0 = -\frac{\pi}{2}$, there are two distinct types of behavior, depending on the value of the exponent $\gamma$. If $\gamma \geq 1$, the wave packet faces an eternal deceleration and only asymptotically reaches the horizon. In contrast,

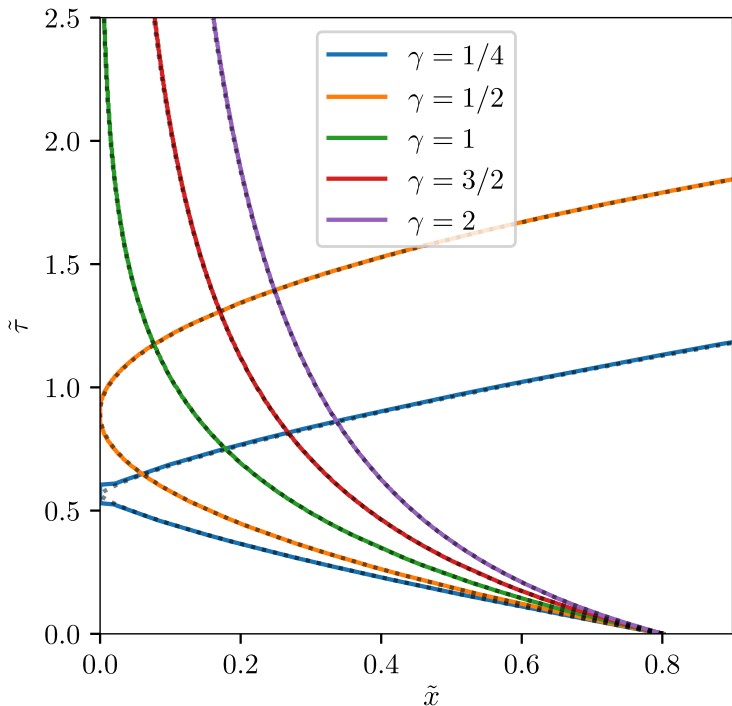

Figure 3: Peak position as a function of time for Gaussian wave packets in the lattice model, for five values of $\gamma$ and using $\tilde{x}_0 = 0.8$, $p_0 = \pi/2$, $\tilde{w} = 0.05$, and lattice of size $N = 1001$. The solid lines represent numerical calculations, while the dashed lines represent the solution of Eq. (37). For $\gamma < 1$, the wave packet bounces on the origin of the lattice, while for $\gamma \geq 1$, it asymptotically reaches the origin of the lattice.

if $\gamma < 1$, the wave packet reaches the point $x = 0$ in a finite time. Although its velocity vanishes there, it does not become stuck. Instead, we observe back-scattering from that point similar to a classical particle bouncing from a hard wall. The value $\gamma = 1$ thus separates a region of parameter space where a wave packet with momentum $p_0 = -\frac{\pi}{2}$ reflects off the origin of the lattice from one where it localizes at the origin. In general relativity, the eternal slowdown of particles is a key feature of particles approaching a black hole horizon as seen by a distant observer. Therefore, the transition at $\gamma = 1$ found here separates lattice models with and without a synthetic horizon. The special case of $\gamma$ being precisely one has previously been shown to mimic a $(1+1)$D anti-de Sitter spacetime [43].

In order to see the effect on the trajectories of changing the initial momentum $p_0$, we show in Fig. 4 the time evolution of the wave packet maximum obtained numerically for $\gamma = 1$ and $\gamma = 1/2$. The numerical results are in good agreement with the semiclassical trajectories given by Eqs. (31) and (32). For the special case of $p_0 = -\pi/2$ and $\gamma = 1$, the wave packet asymptotically approaches the horizon at $x = 0$ at large times $\tau$. For all other momenta, the wave packet bounces back at a nonzero distance $x_{\min} = x_0 \cos p_0$ from the horizon (see also Fig. 5). For $\gamma = 1/2$, the wave packet never localizes at the point $x = 0$, although the velocity vanishes there momentarily for $p_0 = -\pi/2$. Instead we always see a back-scattering from the point $x_{\min} = \sqrt{x_0} \cos p_0$ (as shown also in Fig. 5).

We now turn to the deviations from the semiclassical picture to quantify its breakdown, as signalled by the disintegration of the wave packets stuck to the horizon in Fig. 1. We define a wave packet $\psi_G(\tau)$ whose position $x(\tau)$ and momentum $p(\tau)$ are given by Eqs. (31) and (22), and whose width follows the same time dependence as $x(t)$. Notice that this wave packet is not a solution to the dynamics, but serves as a reference or idealized case to compare

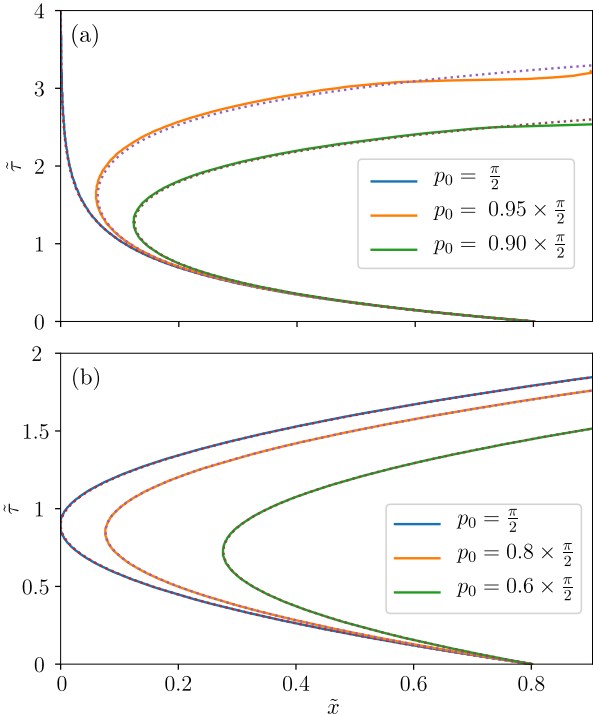

Figure 4: Peak position of Gaussian wave packets as a function of time for three values of $p_0$, and for (a) $\gamma = 1$ and (b) $\gamma = 1/2$, with $\tilde{x}_0 = 0.8$, $\tilde{w} = 0.05$, and $N = 1001$. The solid lines represent the numerical results, while the dashed lines represent the results obtained from the semiclassical Eqs. (31) and (32), for $\gamma = 1$ and $\gamma = 1/2$ respectively.

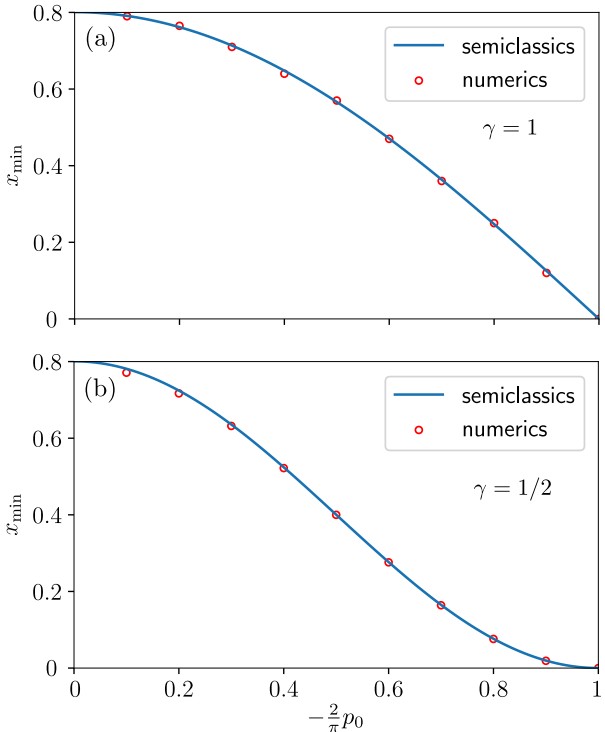

Figure 5: The turning point $x_{\min}$ as a function of the initial momentum $p_0$ for power-law hopping with (a) $\gamma = 1$ and (b) $\gamma = 1/2$. Red circles are the numerical results while solid lines indicate the semiclassical expression $x_{\min} = x_0 \cos^{1/\gamma} p_0$.

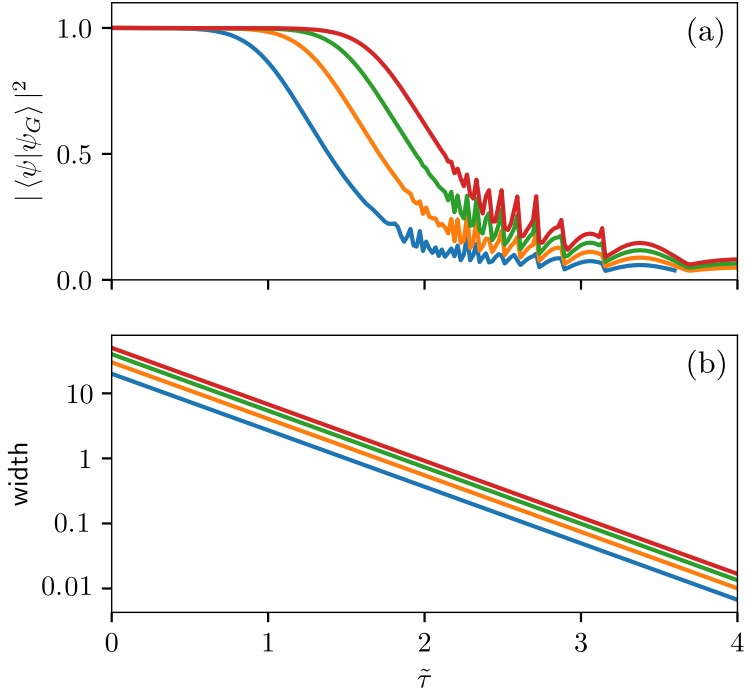

Figure 6: Time evolution of (a) the overlap between $\psi_G(\tau)$ and $\psi(\tau)$ and (b) the wave packet width for four different values of initial width. The drop in overlap coincides with the width of $\psi_G(\tau)$ becoming of the same order as the lattice spacing.

the actual dynamics of a wave packet on the discrete lattice to, for the case of linear hopping. The numerical overlap between these two wave packets is shown as a function of time in Fig. 6(a). It is practically constant and equal to one up to the point where ripples start to appear in the time-evolved wave function, at which point the overlap starts to decrease. The large oscillations seen at late times can be explained by the evolution of $\psi_G$ alone: at large $t$, the width of $\psi_G$ is smaller than the lattice size, and therefore the norm of $\psi_G$ evaluated on the discrete lattice oscillates, depending on whether its peak position is on a lattice point or between sites.

To further analyse the relation between the non-zero lattice spacing and the decreasing overlap and formation of ripples, Fig. 6 shows the overlap between $\psi_G(t)$ and $\psi(t)$ for multiple values of the width. The onset time for the decrease of the overlap goes up with increasing width of the initial wave packet. For initial widths of 20, 30, 40, and 50 lattice spacings, the width of $\psi_G$ when the overlap reaches $1/2$ is 1.31, 0.96, 0.86, and 0.71 lattice spacings respectively. These values are close to the lattice spacing which therefore acts as an effective critical value of the width. We therefore argue that the lattice plays a key role in the formation of the ripples, which are the main observable difference between the dynamics of wave packets with $p_0 = -\pi/2$ in the lattice model and those in relativistic continuum theories.

# 5 Eigenstates and their localization

So far, we studied the dynamics of wave packets in position-dependent lattice models and compared them to a semiclassical picture to highlight possible gravitational analogies. It has also been previously shown that the eternal slowdown of zero-energy wave packets upon approaching the horizon is always associated with the presence of a divergent density of states (DOS) at zero energy, in the $N \to \infty$ limit [43, 47]. In particular, it was pointed out that

the transition from back-scattering to slowdown at $\gamma = 1$ coincides with a transition in the spectral properties of the lattice Hamiltonian at zero energy. In this part, we therefore turn to the eigenstates of the position-dependent lattice models whose properties can shed further light on the features found in the wave packet dynamics as well as the density of states.

The Hamiltonian of Eq. (1) for power law hopping can be written as an $N \times N$ matrix with non-vanishing elements

$$\mathcal{H}_{n,n+1} = -\left(\frac{n}{N-1}\right)^{\gamma}, \quad \mathcal{H}_{n,n-1} = -\left(\frac{n-1}{N-1}\right)^{\gamma}.$$

Then the eigenvalue problem $H|\Psi_\varepsilon\rangle = \varepsilon|\Psi_\varepsilon\rangle$ with $|\Psi_\varepsilon\rangle = (\psi_1, \cdots, \psi_N)^T$ can be written as a set of $N$ coupled equations in the recursive form,

$$-\varepsilon\psi_n = \left(\frac{n-1}{N-1}\right)^{\gamma}\psi_{n-1} + \left(\frac{n}{N-1}\right)^{\gamma}\psi_{n+1}, \tag{38}$$

with the boundary conditions $\psi_0 = \psi_N = 0$.

## 5.1 Exact form for zero-energy states

For the special case of $\varepsilon = 0$ (zero modes), we can find the exact form of the discrete wave-function amplitudes $\psi_n$. Eq. (38) for zero-energy states becomes

$$(n-1)^{\gamma}\psi_{n-1} + n^{\gamma}\psi_{n+1} = 0, \tag{39}$$

which yields

$$\begin{aligned}
\psi_{n+1} &= -\left(\frac{n-1}{n}\right)^{\gamma}\psi_{n-1} \\
&= \left(\frac{n-1}{n}\right)^{\gamma}\left(\frac{n-3}{n-2}\right)^{\gamma}\psi_{n-3} = \cdots.
\end{aligned} \tag{40}$$

Repeating the sequence above, we obtain the wave function amplitudes on even and odd sites

$$\psi_{2n+1} = (-1)^n \frac{[(2n)!]^{\gamma}}{(2^n n!)^{2\gamma}}\psi_1, \tag{41}$$

$$\psi_{2n} = (-1)^{n-1} \frac{[2^n(n-1)!]^{2\gamma}}{[(2n-1)!]^{\gamma}}\psi_2. \tag{42}$$

Equation (39) for $n = 1$ readily shows that $\psi_2 = 0$ for zero-energy states, and, subsequently, from Eq. (42) we find that the wave function amplitude identically vanishes on all even sites.

For an even number of lattice points $N_e = 2N'$, the boundary condition at the second end will read $\psi_{N_e+1} = \psi_{2N'+1} = 0$ which cannot be fulfilled unless all odd lattice points have vanishing amplitudes. As a result, for an even number of lattice points there is no zero mode at all. For an odd number of lattice points on the other hand, the boundary conditions on both ends of the chain force the amplitude to vanish on even lattice points. Therefore for odd $N_o = 2N' + 1$ we find a single zero mode with the wave function given by Eq. (41).

The qualitative behavior of this wave function in the limit of large $n$, can be found using the Stirling's approximation formula $n! \approx \sqrt{2\pi n}(n/e)^n$, with Euler's constant $e$, to be

$$\psi_{2n+1} \approx \frac{(-1)^n}{(\pi n)^{\frac{\gamma}{2}}}\psi_1, \quad n \gg 1. \tag{43}$$

This form applies to the tail of the zero-energy wave function in a large lattice ($N \gg 1$).

## 5.2 Analytical approximation for all eigenstates

Inspired by the approximate power-law form for the zero mode in Eq. (43), we consider a trial solution

$$\psi_n \approx \frac{i^{n-1}}{(\pi n)^{\frac{\gamma}{2}}} e^{i\omega_n}, \qquad n \gg 1, \tag{44}$$

for other eigenstates. This form coincides with the zero-energy eigenstate if $\omega_n = 0$. Substituting this ansatz into the recursive formula of Eq. (38), yields the condition

$$-i\tilde{\varepsilon}\, e^{i\omega_n} = \left[n(n-1)\right]^{\gamma/2} e^{i\omega_{n-1}} - n^\gamma \left(\frac{n}{n+1}\right)^{\gamma/2} e^{i\omega_{n+1}},$$

where $\tilde{\varepsilon} = \varepsilon(N-1)^\gamma$. Approximating $n \pm 1$ with $n$ (for $n \gg 1$) in the prefactors of the exponential then yields

$$-i\frac{\tilde{\varepsilon}}{n^\gamma} = e^{-i(\omega_n - \omega_{n-1})} - e^{i(\omega_{n+1} - \omega_n)}. \tag{45}$$

Finally approximating $\omega_n - \omega_{n-1} \approx \omega_{n+1} - \omega_n \approx d\omega/dx$, with $\omega(x)$ a continuous function such that $\omega(x = n) \equiv \omega_n$, we obtain the differential equation

$$\frac{\tilde{\varepsilon}}{2x^\gamma} = \sin\left(\frac{d\omega}{dx}\right), \tag{46}$$

or equivalently

$$\frac{d\omega}{dx} = i \ln\left[\frac{i\tilde{\varepsilon}}{2x^\gamma} \pm \sqrt{1 - \left(\frac{\tilde{\varepsilon}}{2x^\gamma}\right)^2}\right]. \tag{47}$$

The general solution of this equation (for $\gamma \neq 1$) reads

$$\omega(x) = \mp\frac{\tilde{\varepsilon}}{2x^\gamma} \Xi(x) + ix \ln\left[\frac{i\tilde{\varepsilon}}{2x^\gamma} \pm \sqrt{1 - \frac{\tilde{\varepsilon}^2}{4x^{2\gamma}}}\right],$$

$$\Xi(x) = \frac{\gamma x}{1-\gamma}\, {}_2F_1\left(\frac{1}{2}, \frac{1}{2} - \frac{1}{2\gamma}; \frac{3}{2} - \frac{1}{2\gamma}; \frac{\tilde{\varepsilon}^2}{4x^{2\gamma}}\right). \tag{48}$$

For the special cases $\gamma = 1$ and $\gamma = 1/2$, this gives the respective approximate eigenstates

$$\psi_n = \frac{\left(n + \sqrt{n^2 - \frac{\tilde{\varepsilon}^2}{4}}\right)^{i|\tilde{\varepsilon}|/2}}{i\sqrt{\pi n}} \left[\frac{-\tilde{\varepsilon}}{2n} + i\,\mathrm{sgn}(\varepsilon)\sqrt{1 - \frac{\tilde{\varepsilon}^2}{4n^2}}\right]^n, \tag{49}$$

$$\psi_n = \frac{e^{i\frac{|\tilde{\varepsilon}|}{4}\sqrt{4n - \tilde{\varepsilon}^2}}}{2(\pi n)^{1/4}} \left[\frac{-\tilde{\varepsilon}}{2\sqrt{n}} + i\,\mathrm{sgn}(\varepsilon)\sqrt{1 - \frac{\tilde{\varepsilon}^2}{4n}}\right]^n, \tag{50}$$

respectively.

It should be noted that although they approximate the exact eigenstates, the approximate wave functions $\psi_n$ are not necessarily normalized or even orthogonal to each other. In addition, there is no limitation on the energy $\varepsilon$, except for the fact that we only get mathematically well-behaved results for an energy range consistent with the exact bandwidth. Barring these unavoidable shortcomings and the appearance of some phase differences between $\psi_n$ and the exact eigenstates for low $n$, we find good agreement between the eigenstates obtained numerically by exact diagonalization and the real part of the approximate analytical results of Eqs. (49) and (50), as shown in Figs. 7 and 8. Both for $\gamma = 1$ and $\gamma = 1/2$, two qualitative features of the wave functions stand out: (i) a power-law localization and (ii) regions of suppressed amplitude. States with energies close to zero have an envelope approaching its maximum at

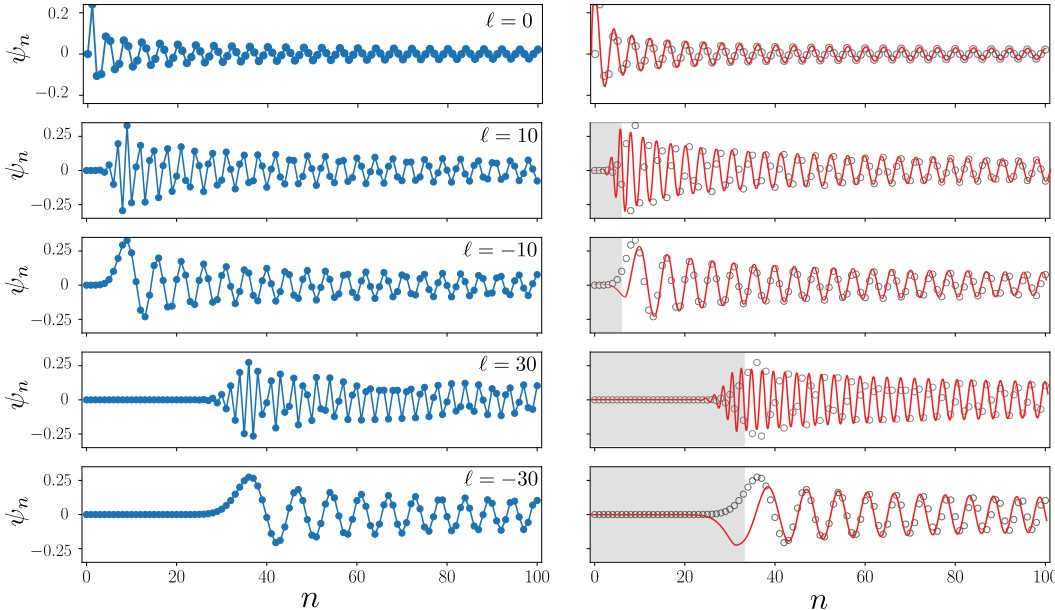

Figure 7: Some of the eigenstates of the lattice model with linearly varying hopping. Left and right panels respectively show the results of exact diagonalization on a lattice with 101 sites, and the corresponding analytical approximation given by Eq. (49). The eigenstates are labeled with integer numbers $\ell$ in the range of $(-[N/2],[N/2])$ such that the zero mode corresponds to $\ell = 0$. The energy of states with $\ell = \pm 10$ and $\ell = \pm 30$, are obtained numerically to be $\varepsilon = \pm 0.14$ and $\varepsilon = \pm 0.67$, respectively. For the sake of comparison, the right panels also display the numerical results with empty circles, indicating the good agreement with the approximate analytical functions shown in red solid lines.

the vicinity of $n = 0$, where the hopping becomes vanishing small. In contrast, the envelope of eigenstates with non-zero energy is suppressed and becomes vanishing small for a finite range around $n = 0$. The extent of the suppressed regions grows with the absolute value of energy, $|\varepsilon_\ell|$, and is bordered by a region with power-law behavior for the envelope of the wave function.

The appearance of forbidden regions is a universal feature for all models with power-law variation of the hopping studied here. This can be understood by noticing that in both Eqs. (46) and (47) the function $\omega(x)$ (or its discrete counterpart $\omega_n$) acquires an imaginary part for $n < n_{c,\varepsilon} \sim (\tilde{\varepsilon}/2)^{1/\gamma}$. As a result, the real part of the trial wave functions in Eq. (44) show an exponentially decaying position-dependence for

$$\frac{n}{N-1} < \frac{n_{c,\varepsilon}}{N-1} \sim \left(\frac{\varepsilon}{2}\right)^{1/\gamma}. \tag{51}$$

These regions appear shaded in the right panels of Figs. 7 and 8.

A more intuitive picture for the existence of forbidden regions can be found by recalling the local band structure picture, with $\varepsilon(n,k) \sim -2[n/(N-1)]^\gamma \cos k$. This shows that for a state with non-zero energy $\varepsilon$, the region with $2[n/(N-1)]^\gamma < |\varepsilon|$ becomes classically forbidden, as there is no available locally extended states there. The only way to penetrate that region is then by quantum tunneling, with its associated exponential decay of wave function amplitudes. Moreover, the existence of forbidden regions for states with non-zero energy provides an alternative explanation for the back-scattering of wave packets with $p_0 \neq -\pi/2$, which have non-zero average energy.

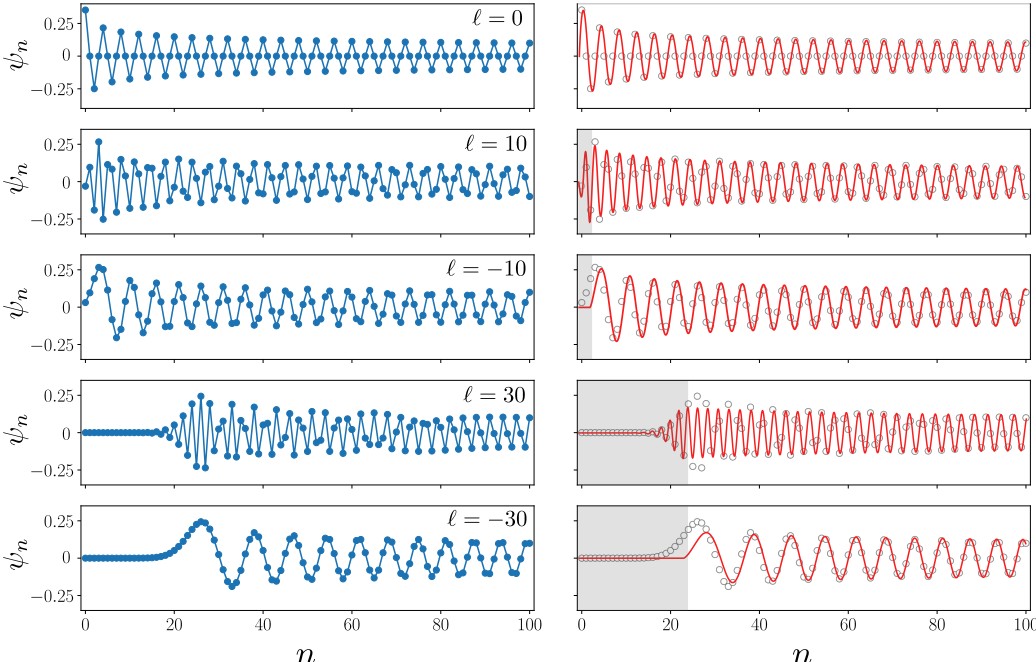

Figure 8: Some of the eigenstates of the lattice model with square-root position dependence of the hopping. Left and right panels respectively show the results of exact diagonalization on a lattice with 101 sites, and the corresponding analytical approximation given by Eq. (49). The eigenstates are labeled with integer numbers $\ell$ in the range of $(-[N/2],[N/2])$ such that the zero mode corresponds to $\ell = 0$. The energy of states with $\ell = \pm 10$ and $\ell = \pm 30$, are obtained numerically to be $\varepsilon = \pm 0.31$ and $\varepsilon = \pm 0.97$, respectively. For the sake of comparison, the right panels also display the numerical results with empty circles, indicating the good agreement with the approximate analytical functions shown in red solid lines.

## 5.3 Exact solution of the lattice model with $\gamma = 1/2$

The lattice model with $t_n = \sqrt{n}$, i.e. $\gamma = 1/2$, is of special interest as one can establish an exact expression for the eigenstates in terms of Hermite polynomials, as we will show in the following. In this case, the eigenvalue equation reads

$$-\tilde{\varepsilon}\psi_n = \sqrt{n-1}\psi_{n-1} + \sqrt{n}\psi_{n+1}\,, \tag{52}$$

with the boundary conditions $\psi_0 = \psi_N = 0$. Below, we show that this recurrence relation can be related to the relation

$$2z\,H_n(z) = 2n\,H_{n-1}(z) + H_{n+1}(z)\,, \tag{53}$$

for the well-known Hermite polynomials

$$H_n(z) = (-1)^n e^{z^2}(\frac{d}{dz})^n e^{-z^2}\,. \tag{54}$$

To show this, let us change to variables $\tilde{\psi}_n$, defined as $\psi_{n+1} = \tilde{\psi}_n/\sqrt{2^n n!}$. The recurrence relation in terms of the new variables then becomes

$$\frac{-\tilde{\varepsilon}\,\tilde{\psi}_{n-1}}{\sqrt{2^{n-1}(n-1)!}} = \frac{\sqrt{n-1}\,\tilde{\psi}_{n-2}}{\sqrt{2^{n-2}(n-2)!}} + \frac{\sqrt{n}\,\tilde{\psi}_n}{\sqrt{2^n n!}}\,, \tag{55}$$

which simplifies to

$$-\sqrt{2}\,\tilde{\varepsilon}\,\tilde{\psi}_n = 2n\,\tilde{\psi}_{n-1} + \tilde{\psi}_{n+1}\,. \tag{56}$$

Comparison of Eqs. (53) and (56) then yields

$$\psi_{n+1} = \frac{\mathcal{A}}{\sqrt{2^n n!}}\,H_n(\frac{-\tilde{\varepsilon}}{\sqrt{2}})\,, \tag{57}$$

with the normalization constant

$$\mathcal{A}_\varepsilon = \frac{1}{N}\,\frac{\sqrt{2^N N!}}{|H_N(\frac{\tilde{\varepsilon}}{\sqrt{2}})|}\,, \tag{58}$$

as derived in detail in appendix B. The boundary condition $\psi_{N+1} = H_N(\frac{\tilde{\varepsilon}}{\sqrt{2}}) = 0$ implies that the eigenvalues $\varepsilon_m$ are directly related to the roots of $N$-th Hermite polynomial. Since $H_N$ is guaranteed to have $N$ real roots, this always yields all $N$ eigenvalues for the problem. In appendix B we provide the asymptotic behavior ($n \gg 1$) of the exact solution found here, and compare it with the corresponding limit of Eq. (50). Unsurprisingly, we find good agreement between the asymptotic forms of the exact and approximate analytical solutions, which can serve as further justification for the method used in Sec. 5.2.

# 6  Summary

We considered a family of nearest-neighbor tight-binding models with position-dependent hopping introduced in [43] whose dynamics for zero-energy wave packets coincide with that of Dirac fields in a static curved spacetime. We extended the results presented in [43] by detailing the numerical simulations of the wave packet dynamics on the lattice and deriving a semiclassical picture valid at long wave lengths, thus further elucidating the analogies between the position-dependent lattice models and a Dirac particle subjected to a gravitational background.

For power-law variation of the hopping with an exponent $\gamma \geq 1$, we showed that zero-energy wave packets with momentum $p_0 = -\pi/2$ eternally slow down while approaching the point of vanishing hopping. This is reminiscent of the dynamics of a particle approaching a black hole horizon as seen by a distant observer. In contrast, for lower values of $\gamma$, the model does not produce horizon physics and wave packets back-scatter from the point of zero hopping. We also showed that wave packets with non-zero average energy, or $p_0 \neq -\pi/2$, never reach the horizon and instead reflect back at a non-zero distance which increases with the absolute value of energy.

To understand the observed wave packet dynamics beyond the semiclassical picture, we studied the eigenstates both numerically and in an approximate analytical way. We found that the low-energy eigenstates have a power-law localization of their wave function envelopes. This behavior of the zero-energy eigenstates results in the formation of ripples and the localization of wave packets observed in the simulated dynamics. The states with non-zero energy, in contrast, show two types of behavior at long and short distances from the horizon. While the long distance behavior is qualitatively similar to low-energy states, at short distances we see exponential localization. The latter comes from the fact that regions with low values for the hopping become classically forbidden for states non-zero energy. These results explain the back-scattering of wave packets with non-zero energy, as they cannot penetrate the forbidden region.

# Acknowledgements

We thank Cosma Fulga, Flavio Nogueira, Lotte Mertens and Viktor Könye for stimulating discussions and acknowledge financial support from the Deutsche Forschungsgemeinschaft (DFG, German Research Foundation), through SFB 1143 project A5 and from the Würzburg-Dresden Cluster of Excellence on Complexity and Topology in Quantum Matter, ct.qmat (EXC 2147, Project Id No. 390858490). A.G.M. acknowledges partial financial support from the Iran Science Elites Federation under Grant No. 11/66332.

# A   Derivation of Dirac equation (7)

The covariant Dirac equation in curved spacetime can be written in the general form

$$(i\underline{\gamma}^{\mu}(x)D_{\mu} - m)\psi = 0\,, \tag{59}$$

with spacetime-dependent Dirac matrices satisfying anticommutation relation $\{\underline{\gamma}^{\mu}(x), \underline{\gamma}^{\nu}(x)\} = 2g^{\mu\nu}(x)$. The covariant derivative for a spinor field is also given by [48],

$$D_{\mu} = \partial_{\mu} + \frac{1}{8}\omega_{\mu ab}[\gamma^{a}, \gamma^{b}]\,, \tag{60}$$

with spin connection components $\omega_{\mu ab}$ and Dirac matrices $\gamma^{a}$ in a flat spacetime. It is convenient to write the contravariant metric as $g^{\mu\nu} = \eta^{ab}e_{a}{}^{\mu}e_{b}{}^{\nu}$, in terms of a vierbein (the local frame field also known as tetrad) with components $e_{a}{}^{\mu}$ and Minkowski metric $\eta^{ab}$, which implies $\underline{\gamma}^{\mu}(x) = \gamma^{a}e_{a}{}^{\mu}$. Accordingly, for the covariant metric we have $ds^{2} = g_{\mu\nu}dx^{\mu}dx^{\nu} = \eta_{ab}e^{a}e^{b}$, using co-vierbeins $e^{a} = e^{a}{}_{\mu}dx^{\mu}$.

The (1+1)D metric (8) can be recast as

$$ds^{2} = -[v(x)d\tau]^{2} + dx^{2} = -(e^{0})^{2} + (e^{1})^{2}\,, \tag{61}$$

which gives the co-vierbeins $e^{0} = v(x)d\tau$ and $e^{1} = dx$ using the convention $\eta_{ab} = \text{diag}(-1, 1)$. We also find vierbeins $e_{0} = \partial_{\tau}/v(x)$ and $e_{1} = \partial_{x}$ simply from $e^{a}e_{b} = \delta^{a}_{b}$. Choosing the representation of gamma matrices $\gamma^{0} = -i\sigma_{x}$ and $\gamma^{1} = \sigma_{y}$, and therefore $[\gamma^{0}, \gamma^{1}] = 2\sigma_{z}$, we only need to find the spin connection components $\omega_{\mu 01} = -\omega_{\mu 10}$. One easy way to calculate them is to use the so-called torsion-free condition [49],

$$de^{a} + \eta^{ab}\omega_{bc} \wedge de^{c} = 0\,, \tag{62}$$

for the spin connection one-form $\omega_{ab} = dx^{\mu}\omega_{\mu ab}$. Considering the co-vierbeins found above, we get $de^{0} = -(dv/dx)d\tau \wedge dx$ and $de^{1} = 0$ which eventually results in $\omega_{01} = -(dv/dx)d\tau$. Therefore, the Dirac Eq. (59) for our curved (1+1)D spacetime reads

$$\left[\frac{\sigma_{x}}{v(x)}\left(\partial_{\tau} - \frac{1}{2}\frac{dv}{dx}\sigma_{z}\right) + i\sigma_{y}\partial_{x} - m\right]\psi = 0\,, \tag{63}$$

which can be re-written as

$$\left[\partial_{\tau} - \sigma_{z}v(x)\partial_{x} - \frac{1}{2}\sigma_{z}\frac{dv}{dx} - mv(x)\sigma_{x}\right]\psi = 0\,, \tag{64}$$

which reduces to Eq. (7) in massless limit ($m = 0$).

# B   Wave functions in the model with $\gamma = 1/2$

In this appendix, we provide supplementary details about the exact solution for the model with square-root position dependence of the hopping.

First, to normalize the wave functions given by Eq. (57), the Christoffel–Darboux formula is used. It applies to sequences of orthogonal polynomials in general, and for Hermite polynomials reads [50],

$$\sum_{n=0}^{N-1}\frac{H_n(z)H_n(z')}{2^n\,n!}=\frac{1}{2^N\,(N-1)!}\,\frac{H_N(z)H_{N-1}(z')-H_{N-1}(z)H_N(z')}{z-z'}\,. \tag{65}$$

For the limit $z' \to z$ this results in

$$\sum_{n=0}^{N-1}\frac{[H_n(z)]^2}{2^n\,n!}=\frac{1}{2^N\,(N-1)!}\left[H'_N(z)H_{N-1}(z)-H'_{N-1}(z)H_N(z)\right]. \tag{66}$$

Using this identity, the normalization factor follows from

$$1=\sum_{n=0}^{N-1}|\psi_n|^2=|\mathcal{A}|^2\sum_{n=0}^{N-1}\frac{[H_n(\frac{\varepsilon}{\sqrt{2}})]^2}{2^n\,n!}$$
$$=|\mathcal{A}|^2\,\frac{H'_N(\frac{\varepsilon}{\sqrt{2}})H_{N-1}(\frac{\varepsilon}{\sqrt{2}})}{2^N\,(N-1)!}\,. \tag{67}$$

This finally results in Eq. (58) upon using the recursion relation $H'_N(z) = 2N\,H_{N-1}(z)$ of Hermite polynomials.

For the particular case of the zero energy state, we obtain

$$\mathcal{A}_{\varepsilon=0}=\frac{\sqrt{N!}}{N!!}\approx\left(\frac{\pi}{2N}\right)^{1/4}, \tag{68}$$

which uses the observation that $H_{N-1}(0) = 2^{(N-1)/2}(N-2)!!$ for odd $N$. For even $N$, $H_{N-1}(0)$ vanishes, and there is no zero mode. To reach the final approximate form, the Stirling formula at large $N$ has been employed.

We can also find an approximate form for the normalization constant using the asymptotic form of Hermite polynomials at large $N$ derived below, which yields

$$\mathcal{A}_\varepsilon\approx\left(\frac{\pi}{2N}\right)^{\frac{1}{4}}\left(1-\frac{\varepsilon^2}{4N+2}\right)^{\frac{1}{4}}e^{-\frac{\varepsilon^2}{4}}\,. \tag{69}$$

This relation is valid for small and intermediate values of $\varepsilon$ as long as $\varepsilon \ll \sqrt{N}$. It shows that for large $N$ and finite energies the normalization factor exponentially decreases with $\varepsilon$.

We next consider the asymptotic ($n \gg 1$) properties of the eigenstates. We invoke the Hermite differential equation

$$\frac{d^2}{dz^2}H_n(z)-2z\frac{d}{dz}H_n(z)+2nH_n(z)=0\,, \tag{70}$$

which shows that for large $n$ but small and intermediate $z$, the Hermite polynomials behave like $\sin(\sqrt{2n}z)$ and $\cos(\sqrt{2n}z)$. Using a more careful analysis, it has been found that [51]

$$e^{-\frac{z^2}{2}}H_n(z)\sim\sqrt{2}\left(\frac{2n}{e}\right)^{\frac{n}{2}}\frac{\cos\left(z\sqrt{2n}-\frac{n\pi}{2}\right)}{\left(1-\frac{z^2}{2n+1}\right)^{1/4}}\,. \tag{71}$$

Substituting this asymptotic form in (57) and using Stirling's formula, we find

$$\psi_{n+1} \sim \frac{1}{N}\left(\frac{N-\tilde{\varepsilon}^2/4}{n-\tilde{\varepsilon}^2/4}\right)^{1/4} \frac{\cos\left(\tilde{\varepsilon}\sqrt{n}+\frac{n\pi}{2}\right)}{\cos\left(\tilde{\varepsilon}\sqrt{N}+\frac{N\pi}{2}\right)}. \tag{72}$$

This provides a good approximation for large $n$, provided that $n \gtrsim n_{c,\varepsilon} = \tilde{\varepsilon}^2/4$. The oscillatory part of this expressions can alternatively be derived by taking the real part of the large $n$ limit of Eq. (50).

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
