# Peer review of "Quantum dynamics in 1D lattice models with synthetic horizons"

_SciPost Physics Core, doi:SciPost Phys. Core 5, 042 (2022)_

## Round 1 · Referee Report · Anonymous (Referee 1) · 2022-1-30

Strengths

  • The paper approaches the problem from several angles, using anayltical and numerical techniques which show excellent agreement and provide a compelling picture.
  • The results point to an interesting connection to one-dimensional horizons in black hole physics.
  • The paper is clearly written.

Weaknesses

  • The paper is a clear follow-up of a previous work, and it does not discuss what is new and what is not, and what is the purpose of writing a new paper.
  • The paper appears to ignore critical graphene literature on the subject.

Report

This paper presents an analysis of 1D tight binding models with power lay decaying hoppings, which are mapped to a 1D dirac equation in the presence of power law decaying velocity. Different analytical and numerical approaches are used to show the emergence of an effective horizon for particle propagation, in analogy to a black hole.

The main problem with this work is that it is a follow-up of a previous work by the same authors (PRR 3, L022022 (2021), Ref. [42] in the manuscript). The current work appears to present very little advancement compared to the previous one. The main conclusions are the same, and even some figures are repeated (Fig 1a is Fig. 2 in Ref. [42]. Fig. 3 is Fig. 3 of Ref. [42], Fig 6a is similar to Fig. 4 of Ref. [42]). Both papers study wave packet dynamics and spectral properties, and different values of gamma in v(x) = x^gamma. The only difference appears to be the presence of more details in the derivations. The motivation for writing a new paper is not clear. In my opinion, this paper cannot meet the threshold for SciPost Phys. because of this.

Disregarding the issues with Ref. [42], in general I find these results interesting, and their interpretation sound. The agreement between numerics and semiclassical results is excellent. The emergence of a horizon is clearly established, making the connection with gravitational physics appealing. I believe it could be suitable for SciPost Phys. Core, answering to some questions an recommendations I outline below.

  • In my opinion, the authors should state very clearly in the introduction what was done before in Ref. [42] and what was the motivation to write this paper. It is critical that the authors do this honestly. Bear the readers in mind: if a reader read Ref. [42], what will be learnt by reading this paper afterwards? The discussion section would also benefit from this. The conclusions of Ref. [42] need not be repeated. The authors may rather discuss what is learnt from the new results.

  • The authors might not be surprised to learn that velocity profiles for the Dirac equation have been extensively considered in the graphene literature. In particular, it would be worthwhile to compare the results of this paper with J. Phys. Condens. Matter 21 095501 (2009), where the gamma=1 case is solved in the continuum (sec. 3.4) and a slightly different tight binding model is presented (appendix A). In this reference, the analytical solution for gamma=1 predicts that wave-packets are always transmitted (when there is a regular lattice on both sides of the v(x) = x region). This appears inconsistent with the formation of a horizon described here. Is the horizon related to the presence of open boundary conditions more than to the power law hopping?

  • Given the several analytical solutions presented, in particular for p =pi/2, is there any physical insight to explain why gamma=1 is the critical case for the emergence of an horizon? This result does not appear characteristic of Dirac fermions, but rather of the metric considered, and its geodesics.

  • Another interesting point is the analytical solution for the eigenstates for gamma=1 and 1/2. Given Ref. [42] presented the DOS for these cases, can one understand the exact form of the DOS peak in the gamma=1 case? Does this analytical solution provide insight on why gamma=1 develops a singularity?

  • A minor point: the effective metric in the continuum Dirac equation in Ref. [42] is different from the one derived here. Footnote [19] explains the coordinate change needed to bring this metric to the diagonal form in the current work. Again, for the benefit of the reader comparing the two works it would be worth mentioning this coordinate change and the relation with the anti de Sitter metric here as well.

  • The authors state that the Dirac equation with v(x) represents a Dirac field in a background metric given by Eq. 7, without providing any derivation or citation. This is a subtle point, as Dirac fermions do not couple to the metric, but to the tetrads or vielbein (see standard book “Quantum fields in curved space” by Birrel and Davis for example). Since a lot of the motivation of this paper is the comparsion with gravitational physics, I think this derivation could be fleshed out in more detail.

  • Why is particle-hole symmetry discussed in Eq. 2? It is not mentioned at all in the rest of the paper.

Requested changes

See Report above.

  • validity: high
  • significance: ok
  • originality: low
  • clarity: high
  • formatting: perfect
  • grammar: perfect

Author:  Corentin Morice  on 2022-04-04  [id 2353]

(in reply to Report 1 on 2022-01-30)

We thank the referees for their constructive feedback on our manuscript. We agree that the manuscript wasn’t explicit enough about its connection to Ref. [42], and have now made this connection entirely clear, both in the introduction and in the conclusion. We reply to the other comments in detail below.

First Referee

1- The authors might not be surprised to learn that velocity profiles for the Dirac equation have been extensively considered in the graphene literature. In particular, it would be worthwhile to compare the results of this paper with J. Phys. Condens. Matter 21 095501 (2009), where the gamma=1 case is solved in the continuum (sec. 3.4) and a slightly different tight binding model is presented (appendix A). In this reference, the analytical solution for gamma=1 predicts that wave-packets are always transmitted (when there is a regular lattice on both sides of the v(x) = x region). This appears inconsistent with the formation of a horizon described here. Is the horizon related to the presence of open boundary conditions more than to the power law hopping?

We thank the referee for pointing out this relevant reference to us. However, it appears to us that it considers a linear variation of the mass, not the tilting, as a function of x. Moreover, wave packets may be transmitted in the Graphene model by scattering to another band. Indeed, unlike the two-band model considered in this reference, our work considers a model a single band.

2- Given the several analytical solutions presented, in particular for p =pi/2, is there any physical insight to explain why gamma=1 is the critical case for the emergence of an horizon? This result does not appear characteristic of Dirac fermions, but rather of the metric considered, and its geodesics.

We agree with the referee that the specificity of gamma=1 originates from the metric. That this value is special can be seen on the one hand from the fact that the metric at gamma=1 case corresponds to Rindler spacetime or uniform acceleration in a general relativistic picture. On the other hand, as discussed in Ref. [42], gamma=1 is a critical value above which the density of states diverges at zero energy in the limit of an infinite lattice. Therefore, the specific value gamma=1 provides a direct relation between the GR and condensed matter perspectives of the model, by observing that the presence of a horizon defined by at least uniform acceleration is also associated with the presence of a sufficient (diverging) density of states available at zero energy.

3- Another interesting point is the analytical solution for the eigenstates for gamma=1 and 1/2. Given Ref. [42] presented the DOS for these cases, can one understand the exact form of the DOS peak in the gamma=1 case? Does this analytical solution provide insight on why gamma=1 develops a singularity?

We thank the referee for bringing up this point. In fact, the mathematical properties of DOS and specially the peaks have been explored in our separate work (Ref. 46). There, we thoroughly explored the DOS of lattice models with general position-dependent hopping, and we prove that the singularity of the DOS for gamma=1 is of a logarithmic form. The analytical solutions presented here for the eigenfunctions cannot be used directly establish the energy dependence of the DOS.

4- A minor point: the effective metric in the continuum Dirac equation in Ref. [42] is different from the one derived here. Footnote [19] explains the coordinate change needed to bring this metric to the diagonal form in the current work. Again, for the benefit of the reader comparing the two works it would be worth mentioning this coordinate change and the relation with the anti de Sitter metric here as well.

We agree with the referee and point this out in the revised version of our manuscript.

5- The authors state that the Dirac equation with v(x) represents a Dirac field in a background metric given by Eq. 7, without providing any derivation or citation. This is a subtle point, as Dirac fermions do not couple to the metric, but to the tetrads or vielbein (see standard book “Quantum fields in curved space” by Birrel and Davis for example). Since a lot of the motivation of this paper is the comparsion with gravitational physics, I think this derivation could be fleshed out in more detail.

We thank the referee for this suggestion. In the revision, we added an appendix for the standard derivation of the Dirac equation (Eq. 6) in the presence of the metric given by Eq. 7.

6- Why is particle-hole symmetry discussed in Eq. 2? It is not mentioned at all in the rest of the paper.

We thank the referee for pointing out our lack of explanation here. The presence of an explicit particle-hole symmetry in the lattice model plays a role in the success of the low-energy effective description in terms of a Dirac field subjected to a background metric. The relativistic Dirac field spectrum necessarily has a particle-hole symmetry and therefore, in lattice models which already possess either particle-hole or chiral symmetry, the low-energy physics is more likely of Dirac form. We added a few lines to the revised manuscript to better explain this connection.

Anonymous on 2022-04-05  [id 2356]

(in reply to Corentin Morice on 2022-04-04 [id 2353])

For clarity for future readers who might want to consult this discussion, and since my answer relates to a very specific point, I will answer here rather than providing a new report. In general I am satisfied with the authors responses to my comments, except in what refers to J. Phys. Condens. Matter 21 095501 (2009) (now Ref [32]). In my opinion a more direct comparison to this work is still important, since they appear to be in direct contradiction.

I must apologize I made a typo in the section reference which might have misled the authors. Indeed, sec. 3.4 deals with a mass profile, while I meant to refer to sec. 3.2, entitled "linear velocity profile with massless particles". This section considers a linear velocity profile in region of length $2\delta$ (see Eq. 29), and in the limit $\delta \rightarrow \infty$ reduces to the $\gamma=1$ case in the authors model (up to a redefinition of the spinor $\psi \rightarrow \sqrt{v(x)}\psi$. Note this is a Dirac equation in 1D with Pauli matrix $\sigma_x$, while the authors use $\sigma_z$ which is also equivalent up to a unitary transformation. Given this, it appears the two papers are dealing with an exactly equivalent continuum model. Yet, Ref. [32] predicts that transmission is 1 for any velocity profile. Section 3.1 deals with a step, and sec. 3.2 then argues that any profile can be built out of small steps and the transmission must remain 1. This is also explicitly checked for the linear profile. Again, this appears inconsistent with the existence of a horizon.

The argument about scattering to another band does not solve the controversy in my opinion. Both models deal with a two component spinor. In the graphene case, the two components arise from the two atoms in the unit cell. In this work, the two components arise from the two values of the momentum used in the low energy expansion. Microscopically these are different origins, but the resulting continuum theories are equivalent up to unitary transformations. Since the statement in disagreement is about the continuum theory, I do not see how the existence of higher bands can play a role.

I encourage the authors to take another look at this issue, and I will submit a final report when this is addressed. I honestly believe this can make this a better paper.

Anonymous on 2022-04-15  [id 2387]

(in reply to Anonymous Comment on 2022-04-05 [id 2356])

We thank the referee for clarifying their thoughts on Ref. [32]. Indeed, in section 3.2, this paper considers a system with position-dependent Fermi velocity applied on a spinor using $\sigma_x$, as made clear in Eqs. (11) and (29).

In our case, we use $\sigma_z$ and have left movers and right movers that are decoupled — this is the correct physical basis if one wants to consider the reflection/transmission of such movers at the horizon. Therefore we are reduced to a 1-band problem, which has strictly $t=0$ at the horizon, so that it is impossible to go through it.

Peres could transform his basis to the left/right mover one and get decoupled first order differential equations, as we do, but he chooses not to and gets two second order differential equations (by substitution). Moreover, the velocity profile in his paper consists of three regions: two with constant velocities (v_+ and v_-) and a third region in between where the velocity goes linearly from v_- to v_+. In addition, it is clear from Eq. (42) that v_- and v_+ have to be of the same sign, so one of them has to be set to zero to have a horizon. When either v_+ or v_- becomes zero, one cannot use Peres' scattering treatment which is also prohibited by Eq. (42) in the paper, because there will be no state available on that side. Therefore when we have a horizon, his analysis and the result of full transmission is no longer valid. We will add a discussion about this in the final revision.

---

## Round 1 · Referee Report · Anonymous (Referee 2) · 2022-2-14

Strengths

  • Comprehensive discussion of the physics of free 1d tight-binding models with power-law dependent hoppings
  • Highlighted correspondence with the continuum massless Dirac equation in an external gravitational field and a detailed discussion of limitations thereof
  • Multiple different approaches to tackle the problem

Weaknesses

  • Lack of clarity about a large overlap in results with the authors' previous publication (PHYSICAL REVIEW RESEARCH 3, L022022 (2021))

Report

In the present paper, a rather detailed discussion of the salient phenomenology of free 1d tight-binding quantum models with position-dependent hoppings is presented. The authors' main motivation stems from an attempt to mimic the massless 1d Dirac equation in an external dilaton gravitational background. They put this idea forward recently in PHYSICAL REVIEW RESEARCH 3, L022022 (2021).

Here are my comments on the manuscript:

-I fully share the concern of another referee about the overlap of this paper with the previous publication (PHYSICAL REVIEW RESEARCH 3, L022022 (2021)). It appears that the authors decided to write a long follow-up on their original letter. In itself, I see no problem with that, but I think the current version of the manuscript does not reflect that properly. I encourage the authors to explain in the introduction section that it is indeed a follow-up paper and present clearly what new results are worked out here.

  • In my opinion, the transformation (2) should not be called a particle-hole symmetry. Any particle-hole symmetry should swap particles and holes of a given vacuum and thus should be anti-unitary. In contrast, the transformation (2) is a unitary transformation, so I would not call it a particle-hole transformation. Also, following another referee, I am curious about the importance of the present symmetry in the manuscript.

After these comments are taken care of, I am happy to recommend the paper for publication in SciPost Core.

  • validity: high
  • significance: good
  • originality: low
  • clarity: high
  • formatting: excellent
  • grammar: good

Author:  Corentin Morice  on 2022-04-04  [id 2354]

(in reply to Report 2 on 2022-02-14)

Second Referee

1- In my opinion, the transformation (2) should not be called a particle-hole symmetry. Any particle-hole symmetry should swap particles and holes of a given vacuum and thus should be anti-unitary. In contrast, the transformation (2) is a unitary transformation, so I would not call it a particle-hole transformation. Also, following another referee, I am curious about the importance of the present symmetry in the manuscript.

We thank the referee for making this important point, and agree this was not sufficiently clear in the original manuscript. As the referee states, any proper particle-hole symmetry (PHS) operator is anti-unitary and is thus always accompanied by complex conjugation {\cal K}. Since the Hamiltonian considered here is real, the complex conjugation does not affect it, and we had represented the PHS by a unitary operator U in the original manuscript. In the revised version we introduce the more precise definition of the PHS operator {\cal P} = {\hat U} {\cal K}, which is anti-unitary and also anti-commutes with the Hamiltonian (because the U operator does). Acting on on wave functions with this form of the PHS operator further illuminate its action on electron and hole states in an intuitive manner.

Finally, as mentioned in response to the first referee, the significance of the PHS symmetry is that it guarantees the possibility of an effective low-energy description by the Dirac equation, which always has a particle-hole symmetric spectrum.

---

## Round 2 · Referee Report · Anonymous (Referee 1) · 2022-4-21

Report

After our interchange in the previous round of refereeing, I am now satisfied with the answer regarding the Peres paper and the existence of the horizon, as well with all the other comments. I thank the authors for their answer and I now recommend publication in SciPost Physics Core, provided the authors include the promised discussion on this matter.

---

## Round 2 · Referee Report · Anonymous (Referee 2) · 2022-4-22

Report

I am happy with the updated manuscript apart from the discussion about the "particle-hole symmetry". The many-body particle-hole operator $\hat P$ anticommutes with the many-body Hamiltonian. In my opinion, symmetries should always be realized by operators that commute with the Hamiltonian in the Fock space. It is true that in the case of the particle-hole symmetry, one naturally ends up with the first-quantized operator that anticommutes with the first-quantized Hamiltonian (this is nicely explained in Sec IIIA of https://arxiv.org/abs/1512.08882). However, to have a particle-hole symmetry, the second-quantized Hamiltonian acting in the Fock space should still commute with the second-quantized particle-hole transformation. So why do the authors call $\hat P$ a symmetry? After this issue is clarified, I will recommend publication in SciPost Core.

---

## Round 2 · Author Response

We thank the referees for their constructive feedback on our manuscript. We agree that the manuscript wasn’t explicit enough about its connection to Ref. [42], and have now made this connection entirely clear, both in the introduction and in the conclusion. We reply to the other comments in detail below.

First Referee

1- The authors might not be surprised to learn that velocity profiles for the Dirac equation have been extensively considered in the graphene literature. In particular, it would be worthwhile to compare the results of this paper with J. Phys. Condens. Matter 21 095501 (2009), where the gamma=1 case is solved in the continuum (sec. 3.4) and a slightly different tight binding model is presented (appendix A). In this reference, the analytical solution for gamma=1 predicts that wave-packets are always transmitted (when there is a regular lattice on both sides of the v(x) = x region). This appears inconsistent with the formation of a horizon described here. Is the horizon related to the presence of open boundary conditions more than to the power law hopping?

We thank the referee for pointing out this relevant reference to us. However, it appears to us that it considers a linear variation of the mass, not the tilting, as a function of x. Moreover, wave packets may be transmitted in the Graphene model by scattering to another band. Indeed, unlike the two-band model considered in this reference, our work considers a model a single band.

2- Given the several analytical solutions presented, in particular for p =pi/2, is there any physical insight to explain why gamma=1 is the critical case for the emergence of an horizon? This result does not appear characteristic of Dirac fermions, but rather of the metric considered, and its geodesics.

We agree with the referee that the specificity of gamma=1 originates from the metric. That this value is special can be seen on the one hand from the fact that the metric at gamma=1 case corresponds to Rindler spacetime or uniform acceleration in a general relativistic picture. On the other hand, as discussed in Ref. [42], gamma=1 is a critical value above which the density of states diverges at zero energy in the limit of an infinite lattice. Therefore, the specific value gamma=1 provides a direct relation between the GR and condensed matter perspectives of the model, by observing that the presence of a horizon defined by at least uniform acceleration is also associated with the presence of a sufficient (diverging) density of states available at zero energy.

3- Another interesting point is the analytical solution for the eigenstates for gamma=1 and 1/2. Given Ref. [42] presented the DOS for these cases, can one understand the exact form of the DOS peak in the gamma=1 case? Does this analytical solution provide insight on why gamma=1 develops a singularity?

We thank the referee for bringing up this point. In fact, the mathematical properties of DOS and specially the peaks have been explored in our separate work (Ref. 46). There, we thoroughly explored the DOS of lattice models with general position-dependent hopping, and we prove that the singularity of the DOS for gamma=1 is of a logarithmic form. The analytical solutions presented here for the eigenfunctions cannot be used directly establish the energy dependence of the DOS.

4- A minor point: the effective metric in the continuum Dirac equation in Ref. [42] is different from the one derived here. Footnote [19] explains the coordinate change needed to bring this metric to the diagonal form in the current work. Again, for the benefit of the reader comparing the two works it would be worth mentioning this coordinate change and the relation with the anti de Sitter metric here as well.

We agree with the referee and point this out in the revised version of our manuscript.

5- The authors state that the Dirac equation with v(x) represents a Dirac field in a background metric given by Eq. 7, without providing any derivation or citation. This is a subtle point, as Dirac fermions do not couple to the metric, but to the tetrads or vielbein (see standard book “Quantum fields in curved space” by Birrel and Davis for example). Since a lot of the motivation of this paper is the comparsion with gravitational physics, I think this derivation could be fleshed out in more detail.

We thank the referee for this suggestion. In the revision, we added an appendix for the standard derivation of the Dirac equation (Eq. 6) in the presence of the metric given by Eq. 7.

6- Why is particle-hole symmetry discussed in Eq. 2? It is not mentioned at all in the rest of the paper.

We thank the referee for pointing out our lack of explanation here. The presence of an explicit particle-hole symmetry in the lattice model plays a role in the success of the low-energy effective description in terms of a Dirac field subjected to a background metric. The relativistic Dirac field spectrum necessarily has a particle-hole symmetry and therefore, in lattice models which already possess either particle-hole or chiral symmetry, the low-energy physics is more likely of Dirac form. We added a few lines to the revised manuscript to better explain this connection.

Second Referee

1- In my opinion, the transformation (2) should not be called a particle-hole symmetry. Any particle-hole symmetry should swap particles and holes of a given vacuum and thus should be anti-unitary. In contrast, the transformation (2) is a unitary transformation, so I would not call it a particle-hole transformation. Also, following another referee, I am curious about the importance of the present symmetry in the manuscript.

We thank the referee for making this important point, and agree this was not sufficiently clear in the original manuscript. As the referee states, any proper particle-hole symmetry (PHS) operator is anti-unitary and is thus always accompanied by complex conjugation {\cal K}. Since the Hamiltonian considered here is real, the complex conjugation does not affect it, and we had represented the PHS by a unitary operator U in the original manuscript. In the revised version we introduce the more precise definition of the PHS operator {\cal P} = {\hat U} {\cal K}, which is anti-unitary and also anti-commutes with the Hamiltonian (because the U operator does). Acting on on wave functions with this form of the PHS operator further illuminate its action on electron and hole states in an intuitive manner.

Finally, as mentioned in response to the first referee, the significance of the PHS symmetry is that it guarantees the possibility of an effective low-energy description by the Dirac equation, which always has a particle-hole symmetric spectrum.

---

## Editorial Decision

published